# Relationship between Joint and Ligament Structures of the Subtalar Joint and Degeneration of the Subtalar Articular Facet

**DOI:** 10.3390/ijerph20043075

**Published:** 2023-02-09

**Authors:** Ryoya Togashi, Mutsuaki Edama, Mayuu Shagawa, Haruki Osanami, Hirotake Yokota, Ryo Hirabayashi, Chie Sekine, Tomonobu Ishigaki, Hiroshi Akuzawa, Yuki Yamada, Taku Toriumi, Ikuo Kageyama

**Affiliations:** 1Institute for Human Movement and Medical Sciences, Niigata University of Health and Welfare, Niigata 950-3198, Japan; 2Department of Anatomy, School of Life Dentistry at Niigata, Nippon Dental University, Niigata 951-1500, Japan

**Keywords:** subtalar joint instability, chronic ankle instability, footprint, osteoarthritis

## Abstract

This study aimed to clarify the relationship between the joint and ligament structures of the subtalar joint and degeneration of the subtalar articular facet. We examined 50 feet from 25 Japanese cadavers. The number of articular facets, joint congruence, and intersecting angles were measured for the joint structure of the subtalar joint, and the footprint areas of the ligament attachments of the cervical ligament, interosseous talocalcaneal ligament (ITCL), and anterior capsular ligament were measured for the ligament structure. Additionally, subtalar joint facets were classified into Degeneration (+) and (−) groups according to degeneration of the talus and calcaneus. No significant relationship was identified between the joint structure of the subtalar joint and degeneration of the subtalar articular facet. In contrast, footprint area of the ITCL was significantly higher in the Degeneration (+) group than in the Degeneration (−) group for the subtalar joint facet. These results suggest that the joint structure of the subtalar joint may not affect degeneration of the subtalar articular facet. Degeneration of the subtalar articular facet may be related to the size of the ITCL.

## 1. Introduction

Recently, subtalar joint instability was reported to occur in 10–80% of patients with chronic lateral ankle instability [1], but the problem is very difficult to diagnose [1,2]. Reasons for this include that the subtalar joint comprises anatomically complex structures, and subtalar joint instability is similar to chronic ankle instability [2,3]. In a report on a mouse model of combined injury to the talocrural and subtalar joints, cartilage degeneration of the talocrural and subtalar joints was observed at 12 weeks after the ligament was cut, suggesting the possibility of progression to osteoarthritis [4]. Elucidation of detailed anatomical findings is therefore imperative to allow accurate diagnosis and evaluation of subtalar joint instability.

Previous studies suggested that nerves, articular facets and ligaments, and muscles are involved in foot stability [5]. The anatomy of the subtalar joint involves multiple bony articular facets and ligaments [6]. Furthermore, since no muscles attach to the talus, movement of the talus is considered dependent on the forces applied to the joint [7]. These joint and ligament structures are thus considered to be highly involved in stability of the subtalar joint.

The relationships between joint structure of the subtalar joint facet (morphology of the articular facet, joint congruence, and intersecting angle) and degeneration of the subtalar articular facet were long investigated. In addition, the presence of degeneration of the subtalar articular facet was considered to indicate that the joint is unstable [8]. Previous studies using bleached bone revealed that a two-facet configuration for the morphology of the articular facet of the calcaneus, which involves continuity between the anterior facet (AF) and middle facet (MF) and isolation of the posterior facet (PF), is prone to degeneration of the articular facet [8,9]. In terms of joint congruence, the foot with congruence of the subtalar joint facet of the talus and calcaneus is considered more stable than the foot with incongruence of the subtalar joint facet of the talus and calcaneus [10]. The intersecting angle is the angle between the AF and MF of the calcaneus. The calcaneus with a two-facet configuration was reported to show a larger intersecting angle and more frequent degeneration of the subtalar articular facet than that with a three-facet configuration [8,9]. Intersecting angles may therefore also be related to degeneration of the subtalar articular facet [8,9]. However, recent reports on bones with residual soft tissue, including cartilage, found significant associations between morphology of the articular facet, joint congruence, or intersecting angle and degeneration of the subtalar articular facet [11], and agreement is lacking among previous studies.

The ligaments involved in the braking of the subtalar joint include the calcaneofibular ligament [12,13], cervical ligament (CL) [14,15], interosseous talocalcaneal ligament (ITCL) [15,16,17], and anterior capsular ligament (ACaL) [15,16,17]. The calcaneofibular ligament is reportedly important in braking the subtalar joint, but the CL, ITCL, and ACaL were not adequately studied. Jotoku et al. [16] considered that the size of the ligament may reflect mechanical demands on the involved structures. Ligamentous structures may thus be greater in area where the joint is unstable and loaded, but the relationship between joint and ligamentous structures of the subtalar joint and degeneration of the subtalar articular facet remains unclear.

The present study aimed to clarify relationships between joint and ligament structures of the subtalar joint and degeneration of the subtalar articular facet. We hypothesized that the joint structure of the subtalar joint facet is independent of the presence or absence of degeneration of the subtalar articular facet and that ligaments are larger in feet with degeneration of the subtalar articular facet.

## 2. Materials and Methods

### 2.1. Cadavers

This investigation examined 50 feet from 25 Japanese fixed cadavers (mean age at death, 79.4 ± 10.8 years; 26 sides from men, 24 sides from women, 25 right sides, and 25 left sides) donated to Nippon Dental University School of Life Dentistry at Niigata. All cadavers were fixed in 10% formalin that was then replaced with alcohol. Inclusion criteria were no history of surgery on the foot or ankle joint. This investigation was conducted with the approval of the Ethics Committee at our institution (approval no. 18867-220720). This study complied with the Declaration of Helsinki and was conducted after informed consent was obtained from all donor families.

### 2.2. Measurement Procedures

The dissection procedure consisted of amputation of the donor foot 10 cm above the ankle joint, followed by dissection of the skin, subcutaneous tissue, inferior extensor retinaculum, and lower leg muscle and tendons from the isolated foot specimen. In addition, to transect the talus at the subtalar joint, ligaments were cut in the following order: calcaneofibular ligament, lateral talocalcaneal ligament, CL, dorsal talonavicular ligament, ACaL, posterior talocalcaneal ligament, deltoid ligament, and ITCL. To disarticulate the calcaneus at the transverse tarsal joint, ligaments were cut in the following order: bifurcated ligament, dorsal calcaneocuboid ligament, plantar calcaneonavicular ligament, short plantar ligament, long plantar ligament, and plantar calcaneocuboid ligament.

Based on previous studies [8,11,18,19,20,21], the morphology of articular facets was classified by the number of articular facets in each talus and calcaneus according to the morphology of AF and MF. A morphology in which AF and MF were continuous and PF was isolated was considered a two-facet configuration, and a morphology in which AF, MF, and PF were isolated was considered a three-facet configuration (Figure 1). Joint congruence was evaluated based on whether the number of articular facets of the talus and calcaneus were the same or different, using the classification of articular facets described above [10]. Presence of the same number of articular facets for the talus and calcaneus was considered as congruence, while any difference was taken as incongruence (Figure 2). The intersecting angle was measured by placing a protractor (stainless steel protractor and angle finder; General Tools, Beijing, China) just above the articular facet of the calcaneus and measuring the angle between the AF and MF. Only a single measurement was taken (Figure 3).

The footprint area of the ligament attachment was used to evaluate ligament structure. Footprint areas were analyzed for CL, ITCL, and ACaL. These ligaments were carefully exposed from the disarticulated talus and calcaneus, and each ligament was detached from each bone. The detached areas were then colored with a pen to identify the footprint area for each ligament attachment. A three-dimensional (3D) scanner (EinScan Pro HD; SHINING 3D, Hangzhou, China) (specifications from the manufacturer: measurement precision, 0.04 mm) was used to create 3D models of the talus and calcaneus. The 3D model data were transferred to Geomagic Freeform 2021 design software (3D SYSTEMS), and curves were drawn on the boundaries of the footprint area using a pen-type device (Touch; 3D SYSTEMS). Afterward, footprint areas on the talar and calcaneal sides were calculated using Rhinoceros7 3D software (McNeel) (Figure 4). Footprint area was taken as the sum of footprint areas of the talus and calcaneus sides, with one measurement taken for each footprint area of each ligament.

Degeneration of the subtalar articular facet of the talus and calcaneus AF, MF, and PF was evaluated with a gross anatomical method based on the classification of Hirose et al. [22]. This method classifies joint facet degeneration into four grades: grade 1, no pathology; grade 2, swelling or fibrillation as a pre-degenerative change; grade 3, fissure or distinct erosion; or grade 4, cartilage defect. In addition, a previous study showed that osteophyte formation is observed in severe joint degeneration [23]. Therefore, in this study, in addition to the aforementioned grades 3 and 4, osteophyte formation was evaluated as a characteristic feature of degeneration of the subtalar articular facet. Feet with degeneration of any of the articular facets of the talus, articular facets of the calcaneus, or degeneration of the talus and calcaneus were assigned to the Degeneration (+) group. Feet without any degeneration of the talus or calcaneus were assigned to the Degeneration (−) group (Figure 5). The number of articular facets, joint congruence, intersecting angle, footprint area, and degeneration of the subtalar articular facet were evaluated and measured by one examiner.

### 2.3. Reliability of Measurements of Intersecting Angle and Footprint Area

The reliabilities of intersecting angle and footprint area measurements were examined for 10 feet from 5 Japanese fixed cadavers (mean age at death, 72.8 ± 15.9 years; 3 sides from men, 2 sides from women, 5 right sides, and 5 left sides) with no history of ankle or ankle joint surgery. Intersecting angles were measured as described above, and re-measurements were taken on the same day as the first measurement. The footprint area was measured by the method described above, and re-measurements were taken the next day or later.

### 2.4. Statistical Analysis

Statistical analyses were performed using IBM SPSS Statistics Version 28.0 (SPSS Japan, Tokyo, Japan). The intraclass correlation coefficient (ICC) was used to examine the reliability, and intra-examiner reliability (1,1) was calculated for each intersecting angle and footprint area.

Relationships between number of articular facets of the talus, joint congruence, and degeneration of the subtalar articular facet were examined using Fisher’s exact test. The relationship between the number of articular facets of the calcaneus and degeneration of the subtalar articular facet was examined using the Chi-square test. After performing the Shapiro–Wilk test and Levene’s test on the intersecting angle and footprint area, Welch’s *t*-test was used for the footprint area of the ACaL, and a two-sample *t*-test was used for the total footprint area of the CL, ITCL, and ACaL in comparisons of the Degeneration (+) and Degeneration (−) groups of the subtalar joint facet. Intersecting angle and footprint areas of CL and ITCL were compared between the Degeneration (+) and Degeneration (−) groups of the subtalar joint facet using the Mann–Whitney U test. The significance level was set at 5%.

Based on each the Fisher’s exact test, chi-square test, Welch’s *t*-test, two-sample *t*-test, and Mann–Whitney U test, effect size was calculated using the fourfold point correlation coefficient (φ) [24], Cohen’s d (d) [24], and rank biserial correlation (r) [25] and categorized as small (0.1), medium (0.3), or large (0.5) for φ and r, while for d it was classified as small (0.2), medium (0.5), or large (0.8) [24].

## 3. Results

### 3.1. Reliability of Measurements of Intersecting Angle and Footprint Area

ICC (1,1) values for measurement of the intersecting angle and footprint area were 0.989 and 0.975, respectively. According to the criteria of Landis et al. [26], an ICC greater than 0.81 reflects “almost perfect” reliability of the measurement. Therefore, the reliability of measurements of intersecting angle and footprint area in this study were almost perfect.

### 3.2. Joint and Ligament Structure and Degeneration of the Subtalar Articular Facet

In terms of the number of articular facets of the talus, 76% showed a two-facet configuration and 24% had a three-facet configuration. Joint congruence was seen in 88%, with 12% incongruent. Mean intersecting angle for the 50 feet was 138.4 ± 8.6° (Table 1). Mean footprint area for the 50 feet was 112.2 ± 43.6 mm^2^ for CL, 48.4 ± 24.3 mm^2^ for ITCL, and 48.3 ± 25.1 mm^2^ for ACaL. Total footprint area of the CL, ITCL, and ACaL was 208.9 ± 68.8 mm^2^ (Table 2). Degeneration of the subtalar articular facet was seen in 40% of feet (Degeneration (+) group), with 60% in the Degeneration (−) group.

### 3.3. Relationship between Joint and Ligament Structures and Degeneration of the Subtalar Articular Facet

In terms of joint structure, no significant differences were seen in number of articular facets of the talus and calcaneus, joint congruence, or intersecting angle between the Degeneration (+) and Degeneration (−) groups (number of articular facets of talus; *p* = 0.191, φ = 0.172, number of articular facets of calcaneus; *p* = 0.186, φ = 0.187, joint congruence; *p* = 0.544, φ= 0.05, and intersecting angle; *p* = 0.692, r = 0.07) (Table 1). In terms of ligament structure, only footprint area of the ITCL was significantly higher in the Degeneration (+) group compared to the Degeneration (−) group (*p* = 0.036, r = 0.297). No significant group differences between Degeneration (+) and Degeneration (−) groups were seen in footprint areas of the CL (*p* = 0.859, r = 0.025), ACaL (*p* = 0.403, d = 0.223), or total footprint area for the CL, ITCL, and ACaL (*p* = 0.929, d = 0.026) (Table 2).

## 4. Discussion

This study examined the relationship between joint and ligament structures of the subtalar joint and degeneration of the subtalar articular facet in Japanese fixed cadavers. To the best of our knowledge, this represents the first study to examine the relationship between joint and ligament structure of the subtalar joint and degeneration of the subtalar articular facet. Two main findings were obtained from this study. First, no significant relationship was apparent between joint structure of the subtalar joint (number of articular facets of the talus and calcaneus, joint congruence, and intersecting angle) and degeneration of the subtalar articular facet. Second, in terms of the ligamentous structure of the subtalar joint, footprint area of the ITCL was significantly higher in the Degeneration (+) group than in the Degeneration (−) group.

The present study did not find any relationship between joint structure of the subtalar joint (number of articular facets of the talus and calcaneus, joint congruence, or intersecting angle) and degeneration of the subtalar articular facet. These results are similar to those described by Kleipool et al. [11], who examined relationships between the number of articular facets, joint congruence, intersecting angle, and degeneration of the subtalar articular facet in bone specimens with residual soft tissue. Kleipool et al. [11] also considered that the degeneration of the subtalar articular facet observed in bleached bone may reflect secondary features of degeneration, while bone with remaining soft tissue may allow detection of early-stage degeneration of the articular cartilage. In this study, articular cartilage degeneration and osteophyte formation were analyzed as degeneration of the subtalar articular facet, which potentially allowed more comprehensive evaluation than in previous studies. In a review article [27] reporting risk factors for osteoarthritis, systemic factors such as sex, ethnicity, genetics, obesity, diet, bone density, and bone mass, and joint-level factors such as bone morphology, muscle strength, joint alignment, occupation, and sports activities, and joint injury were reported as risk factors for osteoarthritis. The degeneration of the subtalar articular facet observed in the present study was potentially induced by influences other than the joint structure of the subtalar joint.

In the present study, among the ligamentous structures of the subtalar joint, only the footprint area of the ITCL showed significantly higher values in the Degeneration (+) group compared to the Degeneration (−) group for the subtalar joint facet. Previous studies suggested that the CL, ITCL, and ACaL in the subtalar joint are each important for stability of the subtalar joint [12,15,16]. In respect of the CL, a previous study suggests that the larger dimensions of the CL and the location more distant from the subtalar joint axis, compared to the ITCL, suggest a more important function of this ligament in the stability of the subtalar joint [15]. The main findings of this study reported that the location of the ITCL located at the center of the subtalar joint, and structural features of ITCL are similar to those of the cruciate ligaments of the knee [28]. The main function of the ITCL was considered to be the connection of the talus and calcaneus and maintenance of the sub-talar joint axis [28,29]. The function of ACaL is not yet clarified in detail. Furthermore, Jotoku et al. [16] consider that the size of the ligament may reflect the result of mechanical demands. Therefore, in unstable joints, it was expected that the ligament structure would adapt to the mechanical loading. In conclusion, the higher ITCL attachment surface area in the degenerated (+) group at the subtalar joint surface was not due to repetitive internal rotation of the subtalar joint, but rather to the adaptation of the ligament structure due to loading that caused malalignment of the talus and calcaneus. The increased load on the subtalar joint surface due to the loading that caused malalignment of the talus and calcaneus potentially caused the degeneration of the subtalar joint surface.

Several limitations to this study need to be kept in mind. The first is that the lifestyles, medical history, and history of sprains for donors remained unknown. In previous studies, osteoarthritis was reported to show many risk factors [27]. Thickening of the anterior talofibular ligament was also shown to occur after a lateral ankle sprain [30]. We therefore cannot rule out the possibility that the lifetime contributions of donor lifestyle, lower extremity alignment, medical history, and history of sprains potentially caused changes in ligament structure and degeneration of the subtalar articular facet. Second, we did not include the calcaneofibular ligament or inferior extensor retinaculum in our study. The calcaneofibular ligament and inferior extensor retinaculum were suggested as potentially important for stability of the subtalar joint [12,31,32]. However, the structure and function of the CL, ITCL, and ACaL in the subtalar joint were not fully investigated. We therefore prioritized evaluation of the CL, ITCL, and ACaL in this study. Future studies should include the calcaneofibular ligament and inferior extensor retinaculum to evaluate, in detail, the structures and functions of the subtalar joint. Third, this study was conducted only on Japanese fixed cadavers. In a previous study, the morphology of the subtalar articular facet was shown to vary markedly between ethnicities [33]. Such differences may have similar effects on ligament structure. The fourth point is that we did not examine the degeneration of the subtalar articular facet from a histological point of view. A previous study classified joint facet degeneration into four grades: grade 1, no pathology; grade 2, swelling or fibrillation as a pre-degenerative change; grade 3, fissure or distinct erosion; or grade 4, cartilage defects [22]. In addition, a previous study showed that osteophyte formation is observed in severe joint degeneration [23]. Therefore, in this study, in addition to the aforementioned grades 3 and 4, osteophyte formation was evaluated as a characteristic feature of degeneration of the subtalar articular facet. However, the method used in this study did not allow for a detailed study of the degeneration of the subtalar articular facet. Therefore, detailed studies including histological methods are needed in the future. The fifth point is that no biomechanical studies were conducted. In the present study, we investigated the relationship between joint and ligament structures of the subtalar joint and degeneration of the subtalar articular facet. Since ancient times, the relationship between morphology of the subtalar articular facet and degeneration of the subtalar articular facet is reported along with the interpretation that the presence of degeneration of the subtalar articular facet indicates joint instability [8]. The same interpretation was used in the present study. However, the relationship between subtalar joint motion and variations in morphology of the subtalar articular facet, as well as detailed function of the ligaments of the subtalar joint, are not yet clear. Detailed investigations using biomechanical methods are clearly needed.

## 5. Conclusions

In this study, no significant relationship was identified between joint structure and the presence of degeneration of the subtalar articular facet. In terms of ligament structure, only footprint area of the ITCL was significantly higher in the Degeneration (+) group than in the Degeneration (−) group of the subtalar joint facet. The present results suggest that the joint structure of the subtalar joint may not affect degeneration of the subtalar articular facet. In the ligament structure of the subtalar joint, the footprint area of the ITCL was significantly larger in the group with degeneration of the subtalar joint facet, suggesting that degeneration of the subtalar joint facet may be related to the size of the ITCL. Further studies using biomechanical methods are needed to clarify the relationship between subtalar joint motion and variations in subtalar articular facet morphology, as well as the detailed functions of the ligaments of the subtalar joints.

## Figures and Tables

**Figure 1 ijerph-20-03075-f001:**
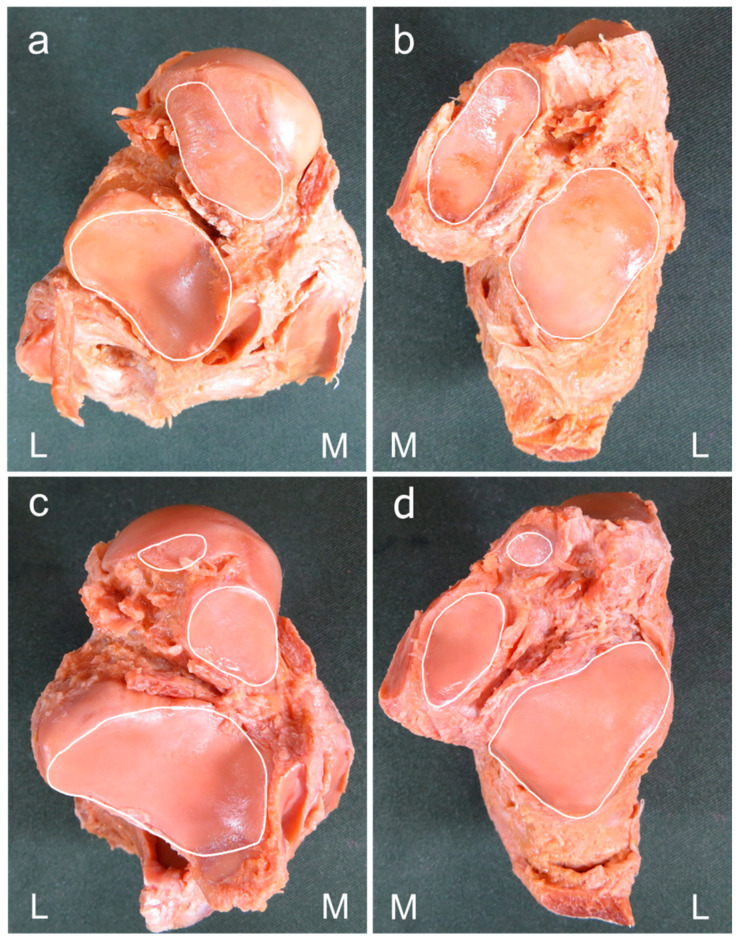
Procedure for classifying the number of subtalar articular facets. (**a**,**c**) Plantar view of the right talus, (**b**,**d**) dorsal view of the right calcaneus. M: medial side; L: lateral side. A morphology in which the anterior facet (AF) and middle facet (MF) are continuous, and the posterior facet (PF) is isolated is considered as a two-facet configuration. A morphology in which AF, MF, and PF are all isolated is considered as a three-facet configuration.

**Figure 2 ijerph-20-03075-f002:**
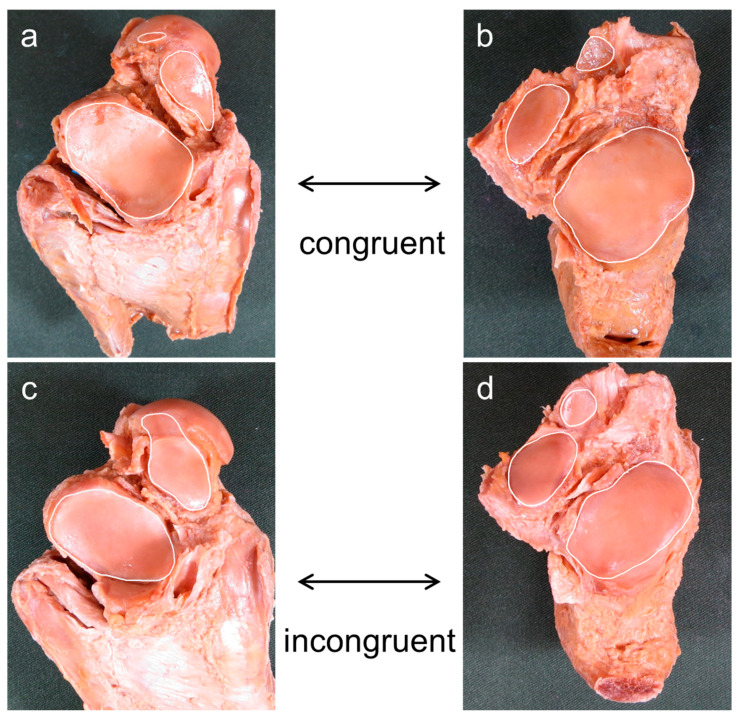
Procedure for assessing joint congruence. (**a**,**c**) Plantar aspect of the right talus; (**b**,**d**) dorsal aspect of the right calcaneus. Joint congruence was evaluated based on whether the number of articular facets of the talus and calcaneus were the same or different based on the classification of the number of the articular facets. The same number of articular facets for both talus and calcaneus were considered to represent congruence (**a**,**b**) and a different number represented incongruence (**c**,**d**).

**Figure 3 ijerph-20-03075-f003:**
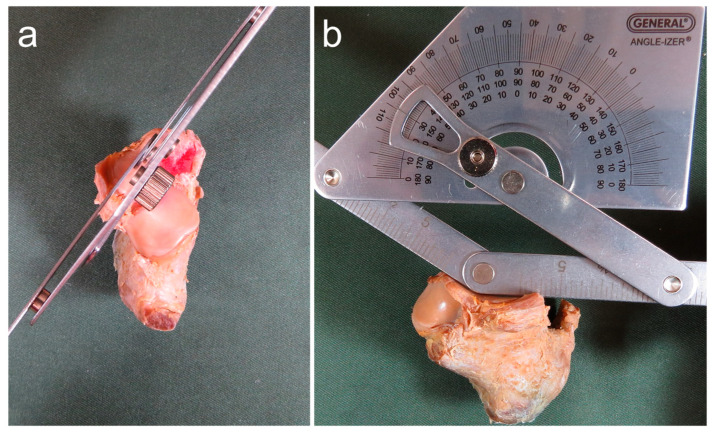
Procedure for measuring the intersecting angle. (**a**) Dorsal view of the right calcaneus; (**b**) Anteromedial view of the right calcaneus. The intersecting angle was measured by placing the protractor just above the articular facet of the calcaneus and measuring the angle between the AF and MF.

**Figure 4 ijerph-20-03075-f004:**
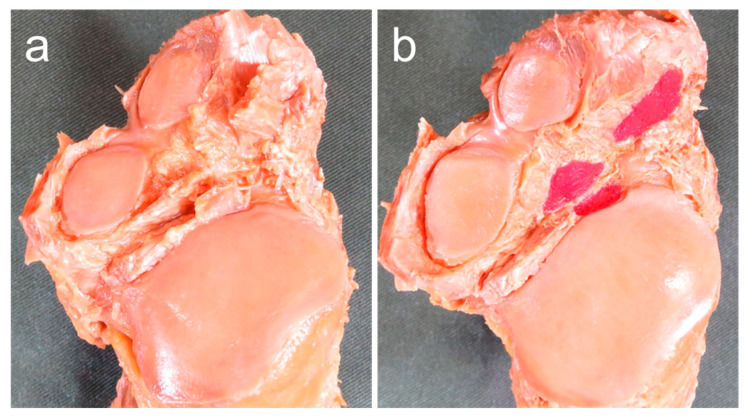
Procedure for measuring footprint area. (**a**) Site of attachment of the cervical ligament, interosseous talocalcaneal ligament, and anterior capsular ligament: right foot, dorsal view; (**b**) after detaching ligaments from bones, detached areas were colored with a pen to identify the footprint for each ligamentous attachment; (**c**) a three-dimensional (3D) scanner was used to create 3D models of the talus and calcaneus; (**d**) a curve was drawn on the boundaries of the footprint using a pen-type device; and (**e**) the footprint area was then calculated using Rhinoceros7 3D software.

**Figure 5 ijerph-20-03075-f005:**
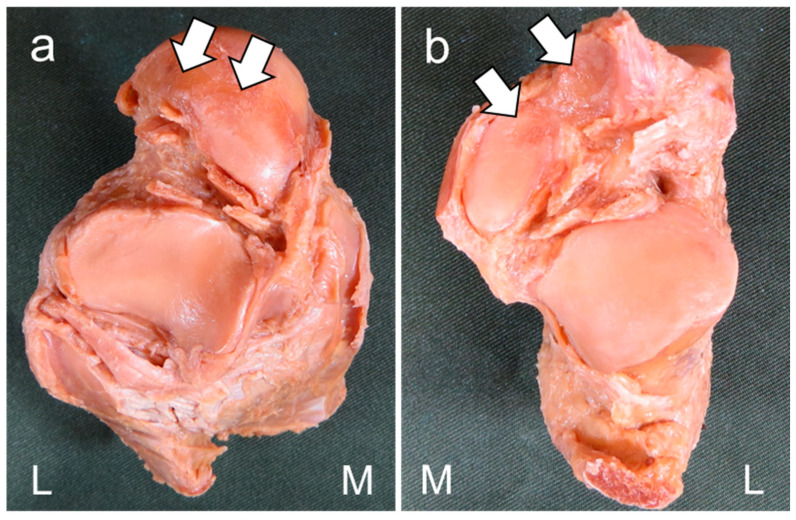
Procedure for assessing degeneration of the subtalar articular facet. (**a**,**c**) Plantar view of the right talus; (**b**,**d**) dorsal view of the calcaneus. arrow: areas showing degeneration of the subtalar articular facet. M: medial side; L: lateral side. Joints showing degeneration of any of the articular facets of the talus, calcaneus, or both were assigned to the Degeneration (+) group. Joints with no degeneration of the talus or calcaneus were assigned to the Degradation (−) group.

**Table 1 ijerph-20-03075-t001:** Relationship between joint structure of the subtalar joint and degeneration of the subtalar articular facet.

	Degeneration		Total
	(+)	(−)	
Number of joint facets (*n*)			
Talar two-facet configuration	17	21	38
Talar three-facet configuration	3	9	12
Calcaneal two-facet configuration	15	17	32
Calcaneal three-facet configuration	5	13	18
Joint congruence (*n*)			
Congruent	18	26	44
Incongruent	2	4	6
Intersecting angle (°)	137.8 ± 6.7	138.7 ± 9.7	138.4 ± 8.6

Congruent: specimens with the same number of articular facets for both talus and calcaneus. Incongruent: specimens with a different number of articular facets for the talus and calcaneus. Degeneration (+): specimen with degeneration of the subtalar articular facet. Degeneration (−): specimens without degeneration of the subtalar articular facet.

**Table 2 ijerph-20-03075-t002:** Relationship between ligament structure of the subtalar joint and degeneration of the subtalar articular facet.

	Degeneration		Total
	(+)	(−)	
Footprint area (mm^2^)			
CL	110.1 ± 46	113.6 ± 42.8	112.2 ± 43.6
ITCL	55.0 ± 21.1	44.1 ± 25.7 *	48.4 ± 24.3
ACaL	44.9 ± 17.9	50.5 ± 29.1	48.3 ± 25.1
Total footprint of CL, ITCL, ACaL	210.0 ± 61.8	208.2 ± 74.2	208.9 ± 68.8

Values are given as mean ± standard deviation. CL: cervical ligament; ITCL: interosseous talocalcaneal ligament; ACaL: anterior capsular ligament. Degeneration (+): specimen with degeneration of the subtalar articular facet. Degeneration (−): specimen without degeneration of the subtalar articular facet. * *p* = 0.036 vs. Degeneration (+) for the subtalar articular facet.

## Data Availability

The datasets used and/or analyzed during the current study are available from the corresponding author on reasonable request.

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
