# Peer review of "Relationship between Joint and Ligament Structures of the Subtalar Joint and Degeneration of the Subtalar Articular Facet"

_ijerph, 2023, doi:10.3390/ijerph20043075_

Round 1

Reviewer 1 Report

Comments as attached:

Author Response

February 6, 2022

Editorial Board

Ref: Manuscript ID: ijerph-2145451

“Relationship between joint and ligament structures of the subtalar

joint and degeneration of the subtalar articular facet” by Mutsuaki Edama

Dear Editor:

Thank you for your letter. We are grateful for the detailed feedback provided by the reviewers, which we feel has helped us to significantly improve the paper. Attached are our point-by-point responses to the reviewers’ comments and our revised manuscript, which we hope will now meet with your approval. For your convenience, we have attached a copy of the manuscript with all revisions highlighted in red font. We believe that our revisions have addressed the issues raised by the reviewers and trust that the manuscript is now suitable for publication in International Journal of Environmental Research and Public Health.

Thank you again for your thoughtful comments, and we look forward to hearing from you soon.

Sincerely,

Mutsuaki Edama

RESPONSE TO REVIEWER #1

  1. This is an inventory study in which only a description of the superficial area of joint surfaces is presented.

→As you pointed out, we have investigated the superficial articular surface of the subtalar joint but we also focused on the evaluation of the presence of the osteophyte. A previous study revealed that the degenerative change of the joint occurred in joint cartilage change first, and in more severe cases, the degenerative change of the joint could identify osteophyte formation [22,23]. Based on the above, we suggest that our investigation could focus on not only superficial points. Because of these things, this study has novelty and defies the boundaries of the previous study sight.

[22] Hirose K, Murakami G, Kura H, Tokita F, Ishii S: Cartilage degeneration in talocrural and talocalcaneal joints from Japanese cadaveric donors. J Orthop Sci, 4: 273-285, 1999.

[23] Kellgren JH, Lawrence JS. Radiological assessment of osteo-arthrosis. Ann Rheum Dis 1957;16(4):494-502, doi:10.1136/ard.16.4.494

  1. A crucial element is the assumed occurrence of degeneration. The mere presence of an irregular surface is taken to indicate degeneration. No direct proof in any way is provided that these surfaces are indeed indicating degeneration. The authors need to provide proof for their suggestion that degeneration has indeed occurred.

→We appreciate your indication. As you mentioned, we could not show certain evidence of degeneration of the subtalar articular facet. Therefore, the discussion has been fixed as suggested. (Discussion Lns 425-444)

The fourth point is that we have not examined the degeneration of the subtalar articular facet from a histological point of view. A previous study has classified joint facet degeneration into four grades: grade 1, no pathology; grade 2, swelling or fibrillation as a pre-degenerative change; grade 3, fissure or distinct erosion; or grade 4, cartilage defects [22]. In addition, a previous study showed that osteophyte formation is observed in severe joint degeneration [23]. Therefore, in this study, in addition to the aforementioned grades 3 and 4, osteophyte formation was evaluated as a characteristic feature of degeneration of the subtalar articular facet. However, the method used in this study did not allow for a detailed study of the degeneration of the subtalar articular facet. Therefore, detailed studies including histological methods are needed in the future. The fifth point is that no biomechanical studies were conducted. In the present study, we investigated the relationship between joint and ligament structures of the subtalar joint and degeneration of the subtalar articular facet. Since ancient times, the relationship between the morphology of the subtalar articular facet and degeneration of the subtalar articular facet has been reported along with the interpretation that the presence of degeneration of the subtalar articular facet indicates joint instability [8]. The same interpretation was used in the present study. However, the relationship between subtalar joint motion and variations in the morphology of the subtalar articular facet, as well as the detailed function of the ligaments of the subtalar joint, are not yet clear. Detailed investigations using biomechanical methods are clearly needed.

[8] Drayer-Verhagen F: Arthritis of the subtalar joint associated with sustentaculum tali facet configuration. J Anat, 183 (Pt 3): 631-634, 1993.

[22] Hirose K, Murakami G, Kura H, Tokita F, Ishii S: Cartilage degeneration in talocrural and talocalcaneal joints from Japanese cadaveric donors. J Orthop Sci, 4: 273-285, 1999.

[23] Kellgren JH, Lawrence JS: Radiological assessment of osteo-arthrosis. Ann Rheum Dis, 16: 494-502, 1957.

  1. The medical history of the persons used is not known, it is impossible to draw any general conclusion.

→In this study, we used no history of surgery on the foot or ankle joint cadavers. We also checked the ligament condition no macroscopic traumatic damage to each ligament when we attempt the dissection of the ligament. Based on the above, we consider that the degeneration of the subtalar articular facet and changing of the interosseous talocalcaneal ligament would be a valid result. However, we consider some medical histories have a possibility that influences the degeneration of the subtalar joint facet and interosseous talocalcaneal ligament’s footprint size, so we described in the discussion part that the degeneration of the subtalar articular facet observed in the present study may thus have been induced by influences other than the joint structure of the subtalar joint (Lns 380-396, 409-415).

(Lns 380-396).

        The present study did not find any relationship between joint structure of the sub-talar joint (number of articular facets of the talus and calcaneus, joint congruence, or in-tersecting angle) and degeneration of the subtalar articular facet. These results were simi-lar to those described by Kleipool et al. [11], who examined relationships between the number of articular facets, joint congruence, intersecting angle and degeneration of the subtalar articular facet in bone specimens with residual soft tissue. Kleipool et al. [11] also considered that the degeneration of the subtalar articular facet observed in bleached bone may reflect secondary features of degeneration, while bone with remaining soft tissue may allow detection of early-stage degeneration of the articular cartilage. In this study, articular cartilage degeneration and osteophyte formation were analyzed as degeneration of the subtalar articular facet, which may have allowed more comprehensive evaluation than in previous studies. In a review article [27] reporting risk factors for osteoarthritis, systemic factors such as sex, ethnicity, genetics, obesity, diet, bone density and bone mass, and joint-level factors such as bone morphology, muscle strength, joint alignment, occupation, and sports activities, and joint injury were reported as risk factors for osteoarthritis. The degeneration of the subtalar articular facet observed in the present study may thus have been induced by influences other than the joint structure of the subtalar joint.

[11]Kleipool, R.P.; Vuurberg, G.; Stufkens, S.A.S.; van der Merwe, A.E.; Oostra, R.-J. Bilateral symmetry of the subtalar joint facets and the relationship between the morphology and osteoarthritic changes. Clinical Anatomy 2020, 33, 997-1006, doi:https://doi.org/10.1002/ca.23525.

[27] Vina, E.R.; Kwoh, C.K. Epidemiology of osteoarthritis: literature update. Curr Opin Rheumatol 2018, 30, 160-167, doi:10.1097/bor.0000000000000479.

    (Lns 409-415).

         Several limitations to this study need to be kept in mind. The first is that the life-styles, medical history, and history of sprains for donors remained unknown. In previous studies, osteoarthritis has been reported to show many risk factors [27]. Thickening of the anterior talofibular ligament has also been shown to occur after a lateral ankle sprain [30]. We therefore cannot rule out the possibility that the lifetime contributions of donor lifestyle, lower extremity alignment, medical history, and history of sprains may have caused changes in ligament structure and degeneration of the subtalar articular facet. Second, we did not include the calcaneofibular ligament or inferior extensor retinaculum in our study. The calcaneofibular ligament and inferior extensor retinaculum have been suggested as potentially important for stability of the subtalar joint [12,31,32]. However, the structure and function of the CL, ITCL, and ACaL in the subtalar joint have not been fully investigated. We therefore prioritized evaluation of the CL, ITCL, and ACaL in this study. Future studies should include the calcaneofibular ligament and inferior extensor retinaculum to evaluate in detail the structures and functions of the subtalar joint. Third, this study was conducted only on Japanese fixed cadavers. In a previous study, the morphology of the subtalar articular facet has been shown to vary markedly between ethnicities [33]. Such differences may have similar effects on ligament structure. The fourth point is that we have not examined the degeneration of the subtalar articular facet from a histological point of view. A previous study has classified joint facet degeneration into four grades: grade 1, no pathology; grade 2, swelling or fibrillation as a pre-degenerative change; grade 3, fissure or distinct erosion; or grade 4, cartilage defects [22]. In addition, a previous study showed that osteophyte formation is observed in severe joint degeneration [23]. Therefore, in this study, in addition to the aforementioned grades 3 and 4, osteophyte formation was evaluated as a characteristic feature of degeneration of the subtalar articular facet. How-ever, the method used in this study did not allow for a detailed study of the degeneration of the subtalar articular facet. Therefore, detailed studies including histological methods are needed in the future. The fifth point is that no biomechanical studies were conducted. In the present study, we investigated the relationship between joint and ligament structures of the subtalar joint and degeneration of the subtalar articular facet. Since ancient times, the relationship between morphology of the subtalar articular facet and degeneration of the subtalar articular facet has been reported along with the interpretation that the presence of degeneration of the subtalar articular facet indicates joint instability [8]. The same interpretation was used in the present study. However, the relationship between subtalar joint motion and variations in morphology of the subtalar articular facet, as well as detailed function of the ligaments of the subtalar joint, are not yet clear. Detailed investigations using biomechanical methods are clearly needed.

   [12] Pellegrini, M.J.; Glisson, R.R.; Wurm, M.; Ousema, P.H.; Romash, M.M.; Nunley, J.A., 2nd; Easley, M.E. Systematic Quantification of Stabilizing Effects of Subtalar Joint Soft-Tissue Constraints in a Novel Cadaveric Model. J Bone Joint Surg Am 2016, 98, 842-848, doi:10.2106/jbjs.15.00948.

  [22] Hirose, K.; Murakami, G.; Kura, H.; Tokita, F.; Ishii, S. Cartilage degeneration in talocrural and talocalcaneal joints from Japanese cadaveric donors. J Orthop Sci 1999, 4, 273-285, doi:10.1007/s007760050104.

   [23] Kellgren, J.H.; Lawrence, J.S. Radiological assessment of osteo-arthrosis. Ann Rheum Dis 1957, 16, 494-502, doi:10.1136/ard.16.4.494.

[27] Vina, E.R.; Kwoh, C.K. Epidemiology of osteoarthritis: literature update. Curr Opin Rheumatol 2018, 30, 160-167, doi:10.1097/bor.0000000000000479.

   [30] Liu, K.; Gustavsen, G.; Royer, T.; Wikstrom, E.A.; Glutting, J.; Kaminski, T.W. Increased ligament thickness in previously sprained ankles as measured by musculoskeletal ultrasound. J Athl Train 2015, 50, 193-198, doi:10.4085/1062-6050-49.3.77.

   [31] Stephens, M.M.; Sammarco, G.J. The stabilizing role of the lateral ligament complex around the ankle and subtalar joints. Foot Ankle 1992, 13, 130-136, doi:10.1177/107110079201300304.

[32] Weindel, S.; Schmidt, R.; Rammelt, S.; Claes, L.; Campe, A.v.; Rein, S. Subtalar instability: a biomechanical cadaver study. Archives of Orthopaedic and Trauma Surgery 2010, 130, 313-319, doi:10.1007/s00402-008-0743-2.

   [33] Bunning, P.S.; Barnett, C.H. A COMPARISON OF ADULT AND FOETAL TALOCALCANEAL ARTICULATIONS. J Anat 1965, 99, 71-76.

  1. Most data are mentioned in the text as well as in Tables. Repetition of the data needs to be avoided.

→Thank you for your indication. Most data are mentioned in the text as well as in Tables, so we have fixed the number in the text as you suggested and have used only percentages. (Result: 3.2, 3.3)

3.2. Joint and ligament structure and degeneration of the subtalar articular facet

In terms of the number of articular facets of the talus, 76% showed a two-facet configuration and 24% had a three-facet configuration. Joint congruence was seen in 88%, with 12% incongruent. Mean intersecting angle for the 50 feet was 138.4 ± 8.6° (Table 1). Mean footprint area for the 50 feet was 112.2 ± 43.6 mm2 for CL, 48.4 ± 24.3 mm2 for ITCL, and 48.3 ± 25.1 mm2 for ACaL. Total footprint area of the CL, ITCL, and ACaL was 208.9 ± 68.8 mm2 (Table 2). Degeneration of the subtalar articular facet was seen in 40% of feet (Degeneration (+) group), with 60% in the Degeneration (-) group.

3.3. Relationship between joint and ligament structures and degeneration of the subtalar articular facet

In terms of joint structure, no significant differences were seen in number of articular facets of the talus and calcaneus, joint congruence, or intersecting angle between the Degeneration (+) and Degeneration (-) groups (number of articular facets of talus; p = 0.191, φ = 0.172, number of articular facets of calcaneus; p = 0.186, φ = 0.187, joint congruence; p = 0.544, φ= 0.05, intersecting angle; p = 0.692, r = 0.07)(Table 1). In terms of ligament structure, only footprint area of the ITCL was significantly higher in the Degeneration (+) group compared to the Degeneration (-) group (p = 0.036, r = 0.297). No significant group differences between Degeneration (+) and Degeneration (-) groups were seen in footprint areas of the CL (p = 0.859, r = 0.025), ACaL (p = 0.403, d = 0.223), or total footprint area for the CL, ITCL, and ACaL (p = 0.929, d = 0.026) (Table 2).

  1. The absence of paragraphs in the text makes it difficult to read.

→We appreciate for your indication. Based on your review, we fixed our text. (Materials and Methods; 2.2. Measurement procedures, 2.4. Statistical analysis)

2.2. Measurement procedures

The dissection procedure consisted of amputation of the donor foot 10 cm above the ankle joint, followed by dissection of the skin, subcutaneous tissue, inferior extensor retinaculum, and lower leg muscle and tendons from the isolated foot specimen. In addition, to transect the talus at the subtalar joint, ligaments were cut in the following order: calcaneofibular ligament, lateral talocalcaneal ligament, CL, dorsal talonavicular ligament, ACaL, posterior talocalcaneal ligament, deltoid ligament, and ITCL. To disarticulate the calcaneus at the transverse tarsal joint, ligaments were cut in the following order: bifurcated ligament, dorsal calcaneocuboid ligament, plantar calcaneonavicular ligament, short plantar ligament, long plantar ligament, and plantar calcaneocuboid ligament.

Based on previous studies [8,11,18–21], the morphology of articular facets was classified by the number of articular facets in each talus and calcaneus according to the morphology of AF and MF. A morphology in which AF and MF were continuous and PF was isolated was considered a two-facet configuration, and a morphology in which AF, MF, and PF were isolated was considered a three-facet configuration (Figure 1). Joint congruence was evaluated based on whether the number of articular facets of the talus and calcaneus were the same or different, using the classification of articular facets described above [10]. Presence of the same number of articular facets for the talus and calcaneus was considered as congruence, while any difference was taken as in-congruence (Figure 2). Intersecting angle was measured by placing a protractor (Stainless Steel Protractor and Angle Finder; General Tools, China) just above the articular facet of the calcaneus and measuring the angle between the AF and MF. Only a single measurement was taken (Figure 3).

The footprint area of the ligament attachment was used to evaluate ligament structure. Footprint areas were analyzed for CL, ITCL, and ACaL. These ligaments were carefully exposed from the disarticulated talus and calcaneus, and each ligament was detached from each bone. The detached areas were then colored with a pen to identify the footprint area for each ligament attachment. A three-dimensional (3D) scanner (EinScan Pro HD; SHINING 3D, Hangzhou, China) (specifications from the manufacturer: measurement precision, 0.04 mm) was used to create 3D models of the talus and calcaneus. The 3D model data were transferred to Geomagic Freeform 2021 design software (3D SYSTEMS), and curves were drawn on the boundaries of the footprint area using a pen-type device (Touch; 3D SYSTEMS). Afterward, footprint areas on the talar and calcaneal sides were calculated using Rhinoceros7 3D software (McNeel) (Figure 4). Footprint area was taken as the sum of footprint areas of the talus and calcaneus sides, with one measurement taken for each footprint area of each ligament.

Degeneration of the subtalar articular facet of the talus and calcaneus AF, MF, and PF was evaluated with a gross anatomical method based on the classification of Hirose et al. [22]. This method classifies joint facet degeneration into four grades: grade 1, no pathology; grade 2, swelling or fibrillation as a pre-degenerative change; grade 3, fissure or distinct erosion; or grade 4, cartilage defect. In addition, a previous study showed that osteophyte formation is observed in severe joint degeneration [23]. Therefore, in this study, in addition to the aforementioned grades 3 and 4, osteophyte formation was evaluated as a characteristic feature of degeneration of the subtalar articular facet. Feet with degeneration of any of the articular facets of the talus, articular facets of the calcaneus, or degeneration of the talus and calcaneus were assigned to the Degeneration (+) group. Feet without any degeneration of the talus or calcaneus were assigned to the Degeneration (-) group (Figure 5). The number of articular facets, joint congruence, intersecting angle, footprint area, and degeneration of the subtalar articular facet were evaluated and measured by one examiner.

2.4. statistical analysis

Statistical analyses were performed using IBM SPSS Statistics Version 28.0 (IBM, NY, USA). The intraclass correlation coefficient (ICC) was used to examine the reliability, and intra-examiner reliability (1,1) was calculated for each intersecting angle and footprint area.

Relationships between number of articular facets of the talus, joint congruence, and degeneration of the subtalar articular facet were examined using Fisher's exact test. The relationship between the number of articular facets of the calcaneus and degeneration of the subtalar articular facet was examined using the Chi-square test. After performing the Shapiro–Wilk test and Levene's test on the intersecting angle and footprint area, Welch's t-test was used for the footprint area of the ACaL, and a two-sample t-test was used for the total footprint area of the CL, ITCL, and ACaL in comparisons of the Degeneration (+) and Degeneration (-) groups of the subtalar joint facet. Intersecting angle and footprint areas of CL and ITCL were compared between the Degeneration (+) and Degeneration (-) groups of the subtalar joint facet using the Mann–Whitney U test. The significance level was set at 5%.

Based on each the Fisher’s exact test, chi-square test, Welch’s t-test, two-sample t-test, and Mann-Whitney U test, effect size was calculated using the fourfold point correlation coefficient (φ) [24], Cohen’s d (d) [24], and rank biserial correlation (r) [25] and categorized as small (0.1), medium (0.3), or large (0.5) for φ and r, while for d it was classified as small (0.2), medium (0.5), or large (0.8)[24].

[8] Drayer-Verhagen F: Arthritis of the subtalar joint associated with sustentaculum tali facet configuration. J Anat, 183 (Pt 3): 631-634, 1993.

[10] Badalahu, Qin B, Luo J, Zeng Y, Fu S, Zhang L: Classification of the subtalar articular surface and its matching situation: an anatomical study on Chinese subtalar joint. Surg Radiol Anat, 42: 1133-1139, 2020.

[11] Kleipool RP, Vuurberg G, Stufkens SAS, van der Merwe AE, Oostra R-J: Bilateral symmetry of the subtalar joint facets and the relationship between the morphology and osteoarthritic changes. Clinical Anatomy, 33: 997-1006, 2020.

[18] Jung M-H, Choi BY, Lee JY, Han CS, Lee JS, Yang YC, Cho BP: Types of subtalar joint facets. Surgical and Radiologic Anatomy, 37: 629-638, 2015.

[19] Bruckner J: Variations in the human subtalar joint. J Orthop Sports Phys Ther, 8: 489-494, 1987.

[20] Nozaki S, Watanabe K, Katayose M: Three-dimensional morphometric analysis of the talus: implication for variations in kinematics of the subtalar joint. Surgical and Radiologic Anatomy, 39: 1097-1106, 2017.

[21] Cho H-J, Kwak D-S, Kim I-B: Analysis of movement axes of the ankle and subtalar joints: Relationship with the articular surfaces of the talus. Proceedings of the Institution of Mechanical Engineers, Part H: Journal of Engineering in Medicine, 228: 1053-1058, 2014.

[22] Hirose K, Murakami G, Kura H, Tokita F, Ishii S: Cartilage degeneration in talocrural and talocalcaneal joints from Japanese cadaveric donors. J Orthop Sci, 4: 273-285, 1999.

[23] Kellgren JH, Lawrence JS: Radiological assessment of osteo-arthrosis. Ann Rheum Dis, 16: 494-502, 1957.

[24] Cohen J. Statistical power analysis for the behavioral sciences. Routledge: 2013.

[25] Fritz CO, Morris PE, Richler JJ. Effect size estimates: current use, calculations, and interpretation. J Exp Psychol Gen 2012;141(1):2-18, doi:10.1037/a0024338

  1. The manuscript needs more clarification is how you calculated the sample size for this study in order to know whether this study is underpowered.

→The text has been corrected as suggested significantly. (Materials and Methods; 2.4. Statistical analysis (Lns 316-320), and Result; 3.3 (Lns 339-348))

2.4. Statistical analysis

Based on each the Fisher’s exact test, chi-square test, Welch’s t-test, two-sample t-test, and Mann-Whitney U test, effect size was calculated using the fourfold point correlation coefficient (φ) [24], Cohen’s d (d) [24], and rank biserial correlation (r) [25] and categorized as small (0.1), medium (0.3), or large (0.5) for φ and r, while for d it was classified as small (0.2), medium (0.5), or large (0.8)[24].

Result

3.3. Relationship between joint and ligament structures and degeneration of the subtalar articular facet

In terms of joint structure, no significant differences were seen in number of articular facets of the talus and calcaneus, joint congruence, or intersecting angle between the Degeneration (+) and Degeneration (-) groups (number of articular facets of talus; p = 0.191, φ = 0.172, number of articular facets of calcaneus; p = 0.186, φ = 0.187, joint congruence; p = 0.544, φ= 0.05, intersecting angle; p = 0.692, r = 0.07)(Table 1). In terms of ligament structure, only footprint area of the ITCL was significantly higher in the Degeneration (+) group compared to the Degeneration (-) group (p = 0.036, r = 0.297). No significant group differences between Degeneration (+) and Degeneration (-) groups were seen in footprint areas of the CL (p = 0.859, r = 0.025), ACaL (p = 0.403, d = 0.223), or total footprint area for the CL, ITCL, and ACaL (p = 0.929, d = 0.026) (Table 2).

[24] Cohen J. Statistical power analysis for the behavioral sciences. Routledge: 2013.

[25] Fritz CO, Morris PE, Richler JJ. Effect size estimates: current use, calculations, and interpretation. J Exp Psychol Gen 2012;141(1):2-18, doi:10.1037/a0024338

Reviewer 2 Report

I would like to congratulate the authors for this well-conducted study. The only thing that needs more clarification is how you calculated the sample size for this study in order to know whether this study is underpowered. 

Author Response

February 6, 2022

Editorial Board

Ref: Manuscript ID: ijerph-2145451

“Relationship between joint and ligament structures of the subtalar

joint and degeneration of the subtalar articular facet” by Mutsuaki Edama

Dear Editor:

Thank you for your letter. We are grateful for the detailed feedback provided by the reviewers, which we feel has helped us to significantly improve the paper. Attached are our point-by-point responses to the reviewers’ comments and our revised manuscript, which we hope will now meet with your approval. For your convenience, we have attached a copy of the manuscript with all revisions highlighted in red font. We believe that our revisions have addressed the issues raised by the reviewers and trust that the manuscript is now suitable for publication in International Journal of Environmental Research and Public Health.

Thank you again for your thoughtful comments, and we look forward to hearing from you soon.

Sincerely,

Mutsuaki Edama

RESPONSE TO REVIEWER #2

  1. The only thing that needs more clarification is how you calculated the sample size for this study in order to know whether this study is underpowered.

→We appreciate for your indication. The text has been corrected as suggested significantly. (Materials and Methods; 2.4. Statistical analysis (Lns 316-320), and Result; 3.3 (Lns 339-348))

2.4. Statistical analysis

Based on each the Fisher’s exact test, chi-square test, Welch’s t-test, two-sample t-test, and Mann-Whitney U test, effect size was calculated using the fourfold point correlation coefficient (φ) [24], Cohen’s d (d) [24], and rank biserial correlation (r) [25] and categorized as small (0.1), medium (0.3), or large (0.5) for φ and r, while for d it was classified as small (0.2), medium (0.5), or large (0.8)[24].

Result

3.3. Relationship between joint and ligament structures and degeneration of the subtalar articular face

In terms of joint structure, no significant differences were seen in number of articular facets of the talus and calcaneus, joint congruence, or intersecting angle between the Degeneration (+) and Degeneration (-) groups (number of articular facets of talus; p = 0.191, φ = 0.172, number of articular facets of calcaneus; p = 0.186, φ = 0.187, joint congruence; p = 0.544, φ= 0.05, intersecting angle; p = 0.692, r = 0.07)(Table 1). In terms of ligament structure, only footprint area of the ITCL was significantly higher in the Degeneration (+) group compared to the Degeneration (-) group (p = 0.036, r = 0.297). No significant group differences between Degeneration (+) and Degeneration (-) groups were seen in footprint areas of the CL (p = 0.859, r = 0.025), ACaL (p = 0.403, d = 0.223), or total footprint area for the CL, ITCL, and ACaL (p = 0.929, d = 0.026) (Table 2).

[24] Cohen J. Statistical power analysis for the behavioral sciences. Routledge: 2013.

[25] Fritz CO, Morris PE, Richler JJ. Effect size estimates: current use, calculations, and interpretation. J Exp Psychol Gen 2012;141(1):2-18, doi:10.1037/a0024338

Reviewer 3 Report

The article Relationship between joint and ligament structures of the subtalar joint and degeneration of the subtalar articular facet is well written and well organized. The research topic is original and addresses an important issue. The purpose of this study was to clarify the relationship between the articular and ligamentous structures of the subtalar joint and the degeneration of the subtalar joint facet. The methodology of the adopted method was described in a concise and clear manner. The results have been discussed in a correct manner allowing for easy interpretation.  The discussion was written in a correct manner. Conclusions are presented in a concise manner and are related to the results obtained. In my opinion, the article can be accepted in its present form, and the results presented can be applied in clinical practice.

Author Response

February 6, 2022

Editorial Board

Ref: Manuscript ID: ijerph-2145451

“Relationship between joint and ligament structures of the subtalar

joint and degeneration of the subtalar articular facet” by Mutsuaki Edama

Dear Editor:

Thank you for your letter. We are grateful for the detailed feedback provided by the reviewers, which we feel has helped us to significantly improve the paper. Attached are our point-by-point responses to the reviewers’ comments and our revised manuscript, which we hope will now meet with your approval. For your convenience, we have attached a copy of the manuscript with all revisions highlighted in red font. We believe that our revisions have addressed the issues raised by the reviewers and trust that the manuscript is now suitable for publication in International Journal of Environmental Research and Public Health.

Thank you again for your thoughtful comments, and we look forward to hearing from you soon.

Sincerely,

Mutsuaki Edama

RESPONSE TO REVIEWER #3

  1. The article Relationship between joint and ligament structures of the subtalar joint and degeneration of the subtalar articular facet is well written and well organized. The research topic is original and addresses an important issue. The purpose of this study was to clarify the relationship between the articular and ligamentous structures of the subtalar joint and the degeneration of the subtalar joint facet. The methodology of the adopted method was described in a concise and clear manner. The results have been discussed in a correct manner allowing for easy interpretation. The discussion was written in a correct manner. Conclusions are presented in a concise manner and are related to the results obtained. In my opinion, the article can be accepted in its present form, and the results presented can be applied in clinical practice.

→We appreciate for your helpful review.

Round 2
